# Prognostic model for survival in patients with neuroendocrine carcinomas of the cervix: SEER database analysis and a single-center retrospective study

**Caixian Yu**[1,☯], **Xiaoliu Wu**[1,☯], **Shao Zhang**[1,☯], **Lan Zhang**[2], **Hongping Zhang**[1], **Hongying Yang**[1], **Min Zhao**[3]*, **Zheng Li**[1]*

1 Department of Gynecologic Oncology, The Third Affiliated Hospital of Kunming Medical University (Yunnan Cancer Hospital/Yunnan Cancer Center), Kunming, Yunnan, PR China, 2 Department of Radiotherapy, The Third Affiliated Hospital of Kunming Medical University (Yunnan Cancer Hospital/Yunnan Cancer Center), Kunming, Yunnan, PR China, 3 Medical Administration Department, The Third Affiliated Hospital of Kunming Medical University (Yunnan Cancer Hospital/Yunnan Cancer Center), Kunming, Yunnan, PR China

☯ These authors contributed equally to this work.
* lizheng@kmmu.edu.cn (ZL); 573532996@qq.com (MZ)

**Data Availability Statement:** All relevant data are within the paper and its Supporting Information files.

## Abstract

### Objective

Neuroendocrine carcinoma of the cervix (NECC) is extremely rare in clinical practice. This study aimed to methodologically analyze the clinicopathological factors associated with NECC patients and to develop a validated survival prediction model.

### Methods

A total of 535 patients diagnosed with NECC between 2004 and 2016 were identified from the Surveillance, Epidemiology and End Results (SEER) database, while 122 patients diagnosed with NECC at Yunnan Cancer Hospital (YCH) from 2006 to 2019 were also recruited. Patients from the SEER database were divided into a training cohort (n = 376) and a validation cohort (n = 159) in a 7:3 ratio for the construction and internal validation of the nomogram. External validation was performed in a cohort at YCH. The Kaplan-Meier method was used for survival analysis, the Log-rank method test was used for univariate analysis of prognostic influences, and the Cox regression model was used for multivariate analysis.

### Results

The 3-year and 5-year overall survival (OS) rates for patients with NECC in SEER were 43.6% and 39.7%, respectively. In the training cohort, multivariate analysis showed independent prognostic factors for NECC patients including race, tumor size, distant metastasis, stage, and chemotherapy (p<0.05). For extended application in other cohorts, a nomogram including four factors without race was subsequently created. The consistency index (C-index) of the nomogram predicting survival was 0.736, which was well-validated in the validation cohorts (0.746 for the internal validation cohort and 0.765 for the external validation

**Funding:** This work was supported by National Natural Science Foundation of China (grant NO. 82360533, 81760469, ZL), Yunnan Fundamental Research Project (grant NO. 202201AT070009, ZL, 202201AY070001-140, SZ), Innovation Team of Molecular Diagnosis and Treatment of Cervical Cancer in Kunming Medical University (grant NO. CXTD201906, ZL), Yunnan Province "Ten Thousand People Plan" (grant NO. YNWR-QNBJ-2019-099, ZL), Reserve Talents of Young and Middle-aged Academic and Technical Leaders in Yunnan Province (grant NO. 2019HB049, ZL). The funders had no role in study design, data collection and analysis, decision to publish, or preparation of the manuscript.

**Competing interests:** The authors have declared that no competing interests exist.

cohort). In both the training and validation cohorts, the 3-year survival rates predicted by the nomogram were comparable to the actual ones. We then succeeded in dividing patients with NECC into high- and low-risk groups concerning OS using the nomogram we developed. Besides, univariate analysis showed that chemotherapy ≥4 cycles may improve the OS of patients at YCH with NECC.

## Conclusion

We successfully constructed a nomogram that precisely predicts the OS for patients with NECC based on the SEER database and a large single-center retrospective cohort. The visualized and practical model can distinguish high-risk patients for recurrence and death who may benefit from clinical trials of boost therapy effectively. We also found that patients who received more than 4 cycles of chemotherapy acquired survival benefits than those who received less than 4 cycles.

## Introduction

Neuroendocrine carcinoma of the cervix (NECC) is a rare subtype of gynecologic malignancy, accounting for only 2% of cervical carcinomas [1, 2]. The most common pathological types are small cell (SCNECC) and large cell (LCNECC) neuroendocrine carcinoma [3], which were known as high-grade NECC, and the incidence of SCNECC was only 0.06 per 100,000 women annually in the U.S., compared with 6.6 and 1.2 for squamous cell carcinoma and adenocarcinoma of the cervix, respectively [4]. Because of its rarity, most previous clinical studies are based on case reports and small cohorts with limited patients [5–9]. Despite its rarity, high-grade NECC is an extremely aggressive carcinoma with an early tendency of distant metastases and invasion of regional lymph nodes at diagnosis [10, 11], which translated to a mean progression-free survival time of 16 months and a mean overall survival time of 40 months after primary treatments [10]. Thus, effective models based on the large cohort that possesses the ability to predict the survival of patients with NECC and clinical trials designed for those with a high risk of recurrence and death are demanding.

The Surveillance, Epidemiology and End Results (SEER) database is maintained by the National Cancer Institute to provide reliable and valuable information on tumor statistics [12]. Chen et al. analyzed clinicopathological factors in 290 patients with SCNECC from the SEER database and compared their 5-year overall survival rates to those with common subtypes of cervical cancers [4], but the study failed to extend to other cohorts. Meanwhile, several recent studies based on single-center retrospective investigations revealed that cycles of chemotherapy, chemotherapy regimens, and FIGO 2018 staging system served as prognostic factors for patients with NECC [13–15], while validated prediction models have not been established yet. Fortunately, during the past decade, the survival rate of cervical cancer has increased [16], and novel models to better predict the survival of patients with cancer, such as nomogram, have been evolving [17]. Hu et al. constructed a nomogram with 938 cases of gastric neuroendocrine neoplasms to predict their clinical outcomes [18], while nomograms for predicting NECC patients' survivorship are still warranted.

In the present study, we enrolled patients with NECC in the SEER database and constructed a nomogram to predict their survival accordingly. Then we collected clinicopathologic data from a large retrospective cohort from Yunnan Cancer Hospital (YCH) and validated the

effectiveness of the nomogram. The visualized and practical nomogram we built succeeded in distinguishing high-risk patients for recurrence and death who may benefit from further clinical trials of boost therapy.

## Materials and methods

### SEER data

Patients' information was retrieved using the SEER database and access was approved (SEERID: 15720- Nov2020). We used the International Statistical Classification of Diseases and Related Health Problems, C53.9 (cervical malignancies) for data download. We subsequently restricted the histology to neuroendocrine tumors according to WHO classification: pure NECC (8012/3 and 8013/3 as large cell carcinoma, 8041/3 as small cell carcinoma, 8042/3 and 8246/3 as neuroendocrine carcinoma, NOS), as well as mixed NECC (8045/3 and 8574/3), and limited timeline to the period of 2004–2016. The exclusion criteria were as follows: (1) benign tumors; (2) unclear staging; (3) unclear ethnicity; and (4) overall survival time less than 1 month. We finally enrolled 535 patients and divided them into two datasets for a ratio of 7:3 randomly as previous studies employed [19, 20]: a training cohort (n = 376) for nomogram building and a validation cohort (n = 159) for internal validation. The information we collected included age, race, histology, tumor size, lymph node metastasis (LNM), distant metastasis, stage (AJCC), chemotherapy, radiation, survival status, and months of survival. Age was cut off at a median value of 45 years. The presence or absence of cause-specific death was the endpoint, and both survival and noncausal-specific death were treated as censored.

### Clinical data

**Patients and data collection.**   A total of 122 patients with pathologically confirmed neuroendocrine carcinoma of the cervix at YCH from 2006-1-1 to 2019-12-30 were retrospectively enrolled. Inclusion criteria were: (1) cervical biopsy confirmed neuroendocrine carcinoma of the cervix, including pure NECC and mixed NECC; (2) the patients received primary treatment at YCH. Exclusion criteria were: (1) patients discharged without any treatment, (2) patients were diagnosed with simultaneous malignant tumors and (3) their clinical and pathological information were incomplete. Patients served as external validation cohort and the clinicopathologic characteristics affecting the 3-year progression-free survival and overall survival were analyzed. This study was approved by the Institutional Research Ethics Committee of YCH (KYLX2022001), and informed consent were obtained from all the patients enrolled.

**Clinicopathologic characteristics.**   The information we collected from the YCH cohort included age, menopausal status, histology, tumor size, stage, stromal invasion, lymph vascular space invasion (LVSI), LNM, distant metastasis, chemotherapy, chemotherapy regimens, chemotherapy cycle, radiotherapy, survival status, recurrence, and months of survival. Age was cut off at a median value of 45 years, and patients were staged or re-staged according to the FIGO 2018 cervical cancer staging system [21]. Stromal invasion, LVSI, and LNM were based on postoperative pathology, thus for patients who received primary chemoradiation, their LNM statuses were labeled as "unknown".

**Treatment.**   Totally 102 patients underwent abdominal or laparoscopic radical hysterectomy and pelvic lymph node dissection, with aortic lymph node dissection in 10 patients. 79 patients received primary surgery, while 23 patients received neoadjuvant chemotherapy or chemoradiation consisting of 1–4 cycles of intravenous chemotherapy and/or 20 Gy/2F of brachytherapy (brachytherapy only, n = 1, chemoradiation, n = 5). Among 102 patients who received surgery, there were 92 patients received adjuvant therapy, including concurrent chemoradiation (n = 51), chemotherapy alone (n = 35), and radiation alone (n = 6). A total of 20

patients received primary chemoradiation, including concurrent chemoradiation (n = 17), chemotherapy alone (n = 2), and radiation alone (n = 1). The chemotherapy regimens were mainly etoposide combined with platinum (EP): etoposide at a dosage of 100 mg/m$^2$ and cisplatin at a dosage of 75 mg/m$^2$ or carboplatin AUC = 4–5, intravenous (IV) drip, repeated every 3 weeks; or paclitaxel combined with platinum (TP): paclitaxel at 175 mg/m$^2$, cisplatin at 75 mg/m$^2$ or carboplatin AUC = 4–5, IV drip, repeated every 3 weeks. The total number of chemotherapy cycles was 2 to 8. Radiation included external-beam radiotherapy (EBRT) alone (n = 20), brachytherapy alone (n = 6), or EBRT combined with brachytherapy (n = 49). EBRT was conducted in doses of 40–50.5 Gy/20-28 F at 1.8–2.0 Gy/dose, frequency 1 time/day, 5 times/week. Target areas included the vaginal stump, parametrium, and pelvic lymph node area. Brachytherapy was conducted in doses of 12–30 Gy/2-5 F at 6.0 Gy/dose, frequency 1 time/week.

**Follow-up.**   Patients were followed up by telephone, outpatient, and inpatient data to reveal their recurrence and survival status. Follow-up procedures consisted of history taken and physical examination, along with imaging as indicated, including CT, MRI, and ultrasound; cervical/vaginal cytology and serum tumor markers had also been done. The time from the date of diagnosis to the first tumor recurrence or the last follow-up visit was defined as progression-free survival (PFS), and the time from the date of diagnosis to death due to the tumor or the last follow-up visit was defined as overall survival (OS). The median follow-up time was 47 months (2.0–92.0 months) as of July 1, 2022, with 10 patients lost follow-up and a follow-up rate of 92.4% (122/132).

## Statistical methods

The data were statistically analyzed using SPSS 25.0. The Kaplan–Meier method was used for the analysis of survival among indicators, and the log-rank test was selected for the univariate analysis of prognostic influences, while the Cox regression model was used for multivariate analysis with the test criterion p value < 0.05. The "caret" package in R version 4.1.1 was randomly assigned as the training cohort and the internal validation cohort at a ratio of 7:3. Based on the results of multivariate analysis of the training set (p-value <0.05), a nomogram was created using the "rms" package. To address whether the proportional hazards (PH) assumptions were met, we employed the Global test (Cox.zph function in R) and Schoenfeld residual graph (ggcoxzph function in R) and found all covariates, as well as the whole model, did meet the PH assumption. The performance of the nomogram was checked by the consistency index (C-index). During the internal and external validation of the nomogram, the scores of each patient in the validation cohort were calculated based on the nomogram using the "coxph" function. Then, the total score was used as a factor in the Cox regression of the cohort, and finally, the C-index was calculated based on the regression analysis. The median scores were used as the threshold to divide the patients into high-risk and low-risk groups. The area under the receiver operating characteristic (ROC) curve (AUC) was also used to assess the performance of the prognostic model. The predicted survival was compared with the actual survival, and calibration plots were generated. A two-sided p value < 0.05 was considered statistically significant.

## Results

### Flow chart of the study and clinicopathologic characteristics as well as survival of patients with NECC in SEER

A total of 535 patients in the SEER database and 122 patients from YCH were divided into three datasets: training cohort (n = 376, all from SEER) for nomogram building, internal

validation cohort (n = 159, all from SEER) and external validation cohort (n = 122, all from YCH) for model validation (Fig 1). Based on the SEER database, the 3-year and 5-year OS rates for patients with NECC were 43.6% and 39.7%, respectively. The clinicopathologic characteristics of patients in SEER are shown in Table 1.

## Univariate and multivariate analysis in the training cohort

The median age of 376 patients in the training cohort was 45 years (21–86 years), and the median survival time was 16 months (1–149 months), while 3-year and 5-year OS rates were 38.9% and 35.2%, respectively. Univariate analysis revealed that age, race, tumor size, distant metastasis, stage, LNM, and chemotherapy all had an impact on the OS of patients with NECC (p value<0.05) (Table 2). Then we performed a multivariate analysis based on the factors above. However, we decided not to put LNM in the multivariate analysis because the variable "stage" (AJCC) here, is containing the information of LNM, and that is why we found a strong collinear relationship between LNM and stage by covariate diagnosis. Besides, there were 62.5% of patients in this cohort (235/376) whose LNM statuses were marked as "unknown". Multivariate analysis revealed that race, tumor size, distant metastasis, stage, and chemotherapy are independent risk, or protective factors affecting the OS of patients with NECC (Table 2). The survival analysis of different clinicopathological factors affecting the OS of NECC patients in the training cohort is shown in S1 Fig.

## Nomogram construction

We then constructed a nomogram to predict 3-year and 5-year OS in patients with NECC using above mentioned five clinicopathological factors based on multivariate analysis: race, tumor size, distant metastasis, stage, and chemotherapy (Fig 2A). Since the factor "race" is not appliable in cohorts outside the U.S., we then constructed another nomogram with four factors (tumor size, distant metastasis, stage, and chemotherapy) without race, and all following analyses were conducted based on the four factors model (Fig 2B). The C-index of this nomogram

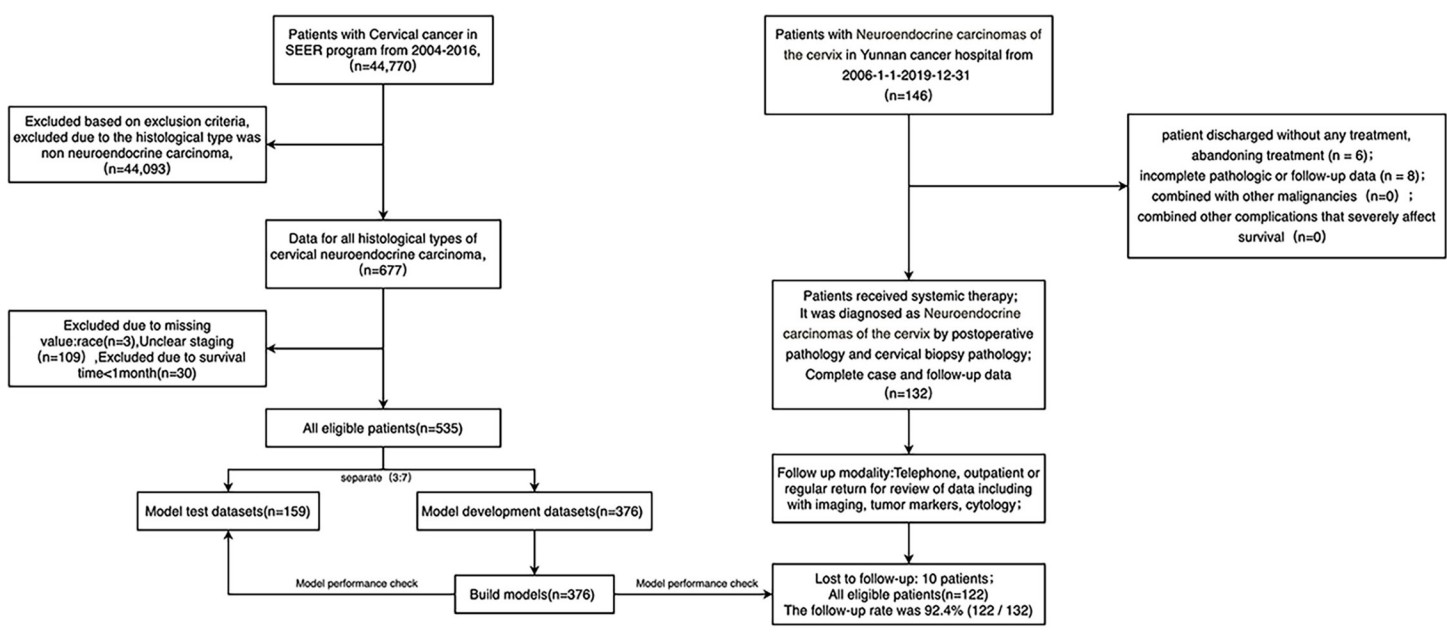

**Fig 1. Flowchart of sample data collection.**

**Table 1. The clinicopathologic characteristics of NECC patients in the training and internal validation cohorts.**

| Variables | | Total cohort (N = 535) | | Training cohort (N = 376) | | Validation cohort (N = 159) | |
|---|---|---|---|---|---|---|---|
| | | n | % | n | % | n | % |
| Age | <45 | 247 | 46.2 | 167 | 44.4 | 80 | 50.3 |
| | ≥45 | 288 | 53.8 | 209 | 55.6 | 79 | 49.7 |
| Race | White | 385 | 72.0 | 266 | 70.7 | 119 | 74.9 |
| | Black | 85 | 15.9 | 61 | 16.2 | 24 | 15.1 |
| | Others | 65 | 12.1 | 49 | 13.1 | 16 | 10.0 |
| Histology | Pure | 496 | 92.7 | 345 | 91.8 | 151 | 95.0 |
| | Mixed | 39 | 7.3 | 31 | 8.2 | 8 | 5.0 |
| Tumor size (cm) | <4 | 129 | 24.1 | 84 | 22.3 | 45 | 28.3 |
| | ≥4 | 251 | 46.9 | 182 | 48.4 | 70 | 44.0 |
| | Unknown | 154 | 29.0 | 110 | 29.3 | 44 | 27.7 |
| Lymph node metastasis | No | 119 | 22.2 | 82 | 21.8 | 37 | 23.3 |
| | Yes | 89 | 16.6 | 59 | 15.7 | 30 | 18.9 |
| | Unknown | 327 | 61.2 | 235 | 62.5 | 92 | 57.8 |
| Distant metastasis | No | 220 | 41.1 | 152 | 40.4 | 68 | 42.7 |
| | Yes | 77 | 14.4 | 53 | 14.1 | 24 | 15.1 |
| | Unknown | 238 | 44.5 | 171 | 45.5 | 67 | 42.2 |
| Stage | I | 147 | 27.5 | 101 | 26.9 | 42 | 26.4 |
| | II | 38 | 7.1 | 31 | 8.2 | 7 | 4.4 |
| | III | 138 | 25.8 | 95 | 25.3 | 43 | 27.0 |
| | IV | 212 | 39.6 | 145 | 38.6 | 67 | 42.2 |
| Chemotherapy | No | 98 | 18.3 | 71 | 18.9 | 27 | 17.0 |
| | Yes | 437 | 81.7 | 305 | 81.1 | 132 | 83.0 |
| Radiation | No | 207 | 38.7 | 134 | 35.6 | 73 | 45.9 |
| | Yes | 328 | 61.3 | 242 | 64.4 | 86 | 54.1 |

was 0.742 (95% CI: 0.726–0.758) and the AUCs for predicting the 3-year and 5-year OS in the training cohort were 0.769 and 0.766, respectively (Fig 3A and 3B), while the predicted 3-year and 5-year survival rates were consistent with the actual ones (Fig 4A and 4B).

## Nomogram validation

In the internal validation cohort, the C-index of this nomogram was 0.733 (95% CI: 0.709–0.757) and the AUCs were 0.73 and 0.73 for 3-year and 5-year OS, respectively (Fig 3C and 3D). Calibration plots also showed the predicted 3-year, and 5-year survival rates were consistent with the actual ones (Fig 4C and 4D). We then validated the model in the external validation cohort. The C-index was 0.792 (95% CI: 0.765–0.819) and the AUCs were 0.832 and 0.734 for 3-year and 5-year OS, respectively (Fig 3E and 3F). In the external validation cohort, calibration plots showed the predicted 3-year OS rates were consistent with the actual ones (Fig 4E), while the predicted 5-year OS rates were far beyond (Fig 4F).

## Effect of clinicopathological factors on PFS and OS in 122 patients with NECC at YCH

The clinicopathological factors of 122 patients with NECC from YCH were characterized as shown in S1 Table. There were 66 recurrences, with a cumulative PFS rate of 53.2% and 35.6% at 3-year and 5-year, respectively, and a median PFS of 25.3 months (0–76.0 months). There were 17 (21.2%) pulmonary metastases, 19 (28.8%) liver metastases, 12 (18.2%) pelvic

**Table 2. Univariate and multivariate analysis of 3-year and 5-year overall survival of NECC patients in the training cohort (N = 376).**

| Variables | | Univariate analysis | | | Multivariate analysis | | |
|---|---|---|---|---|---|---|---|
| | | 3-year OS % | 5-year OS % | P-value | HR | 95% CI | P-value |
| Age | <45 | 45.2 | 42.0 | **0.003**[*] | reference | | 0.876 |
| | ≥45 | 33.3 | 29.0 | | 1.023 | 0.767–1.366 | |
| Race | White | 45.5 | 41.5 | **<0.001**[*] | reference | | <0.001[*] |
| | Black | 19.7 | 14.4 | | 1.876 | 1.311–2.687 | 0.001 |
| | Other | 28.4 | 24.3 | | 1.745 | 1.184–2.570 | 0.005 |
| Histology | Pure | 37.9 | 33.8 | 0.266 | | | |
| | Mixed | 48.0 | 43.2 | | | | |
| Tumor size (cm) | <4 | 54.7 | 54.7 | **<0.001**[*] | reference | | 0.018[*] |
| | ≥4 | 37.9 | 33.2 | | 1.445 | 0.945–2.207 | 0.089 |
| | Unknown | 27.7 | 19.8 | | 1.868 | 1.208–2.887 | 0.005 |
| Lymph node metastasis | No | 58.6 | 56.3 | **<0.001**[*] | | | |
| | Yes | 50 | 44.9 | | | | |
| | Unknown | 27.8 | 22.8 | | | | |
| Distant metastasis | No | 52.5 | 48 | **<0.001**[*] | reference | | 0.031[*] |
| | Yes | 6.7 | 0 | | 1.804 | 1.159–2.809 | 0.009 |
| | Unknown | 37.1 | 33.6 | | 1.276 | 0.929–1.753 | 0.132 |
| Stage | I | 66.9 | 62.1 | **<0.001**[*] | reference | | <0.001[*] |
| | II | 38.1 | 19.1 | | 1.466 | 0.780–2.756 | 0.234 |
| | III | 40.2 | 35.0 | | 2.377 | 1.495–3.782 | <0.001 |
| | IV | 15.4 | 13.8 | | 4.300 | 2.736–6.758 | <0.001 |
| Chemotherapy | No | 33.9 | 28.2 | **0.002**[*] | reference | | <0.001[*] |
| | Yes | 40.2 | 35.7 | | 0.513 | 0.360–0.731 | <0.001 |
| Radiation | No | 41.8 | 40.3 | 0.905 | | | |
| | Yes | 37.4 | 32.6 | | | | |

Abbreviations: OS, Overall Survival; HR, Hazard Ratio, CI, Confidence Interval.

[*]P-value<0.05. Border character indicating parameters included in multivariate analysis. P-values were calculated by the Chi-square test or Fisher's exact test.

recurrences, 4 brain metastases (6.06%), 5 bone metastases (7.58%) and 9 multisite metastases (13.6%, ≥2 metastases). At the end of follow-up, there were 56 deaths, with 3-year and 5-year OS rates of 66.4% and 35.7%, respectively, and a median OS of 33.0 months (2.1–84.0 months). Univariate analysis showed high serum NSE level, tumor size, stage, deep 1/3 stromal invasion, LVSI, LNM, and distant metastasis may reduce PFS in NECC patients (Table 3). Since LNM was referred to as stage IIIC according to the FIGO 2018 staging system, we included stage but not LNM in multivariate analysis and found tumor size, deep 1/3 stromal invasion, and stage were independent prognostic factors affecting PFS in NECC patients from YCH. The survival analysis of different clinicopathological factors affecting the PFS of NECC patients is shown in S2 Fig. Moreover, univariate analysis showed that a total number of chemotherapies ≥4 cycles was associated with improvement OS in NECC patients, and multivariate analysis demonstrated that tumor size, stage, deep 1/3 stromal invasion, distant metastasis, and cycles of chemotherapy were independent prognostic factors affecting OS in NECC patients (Table 4). Our study also analyzed the prognostic impact of five immunohistochemical markers including Syn, CgA, NSE, CD5, and KI-67. Except that Syn negativity led to a poorer prognosis, none of the other markers showed prognostic value in the multivariate model (S2 Table).

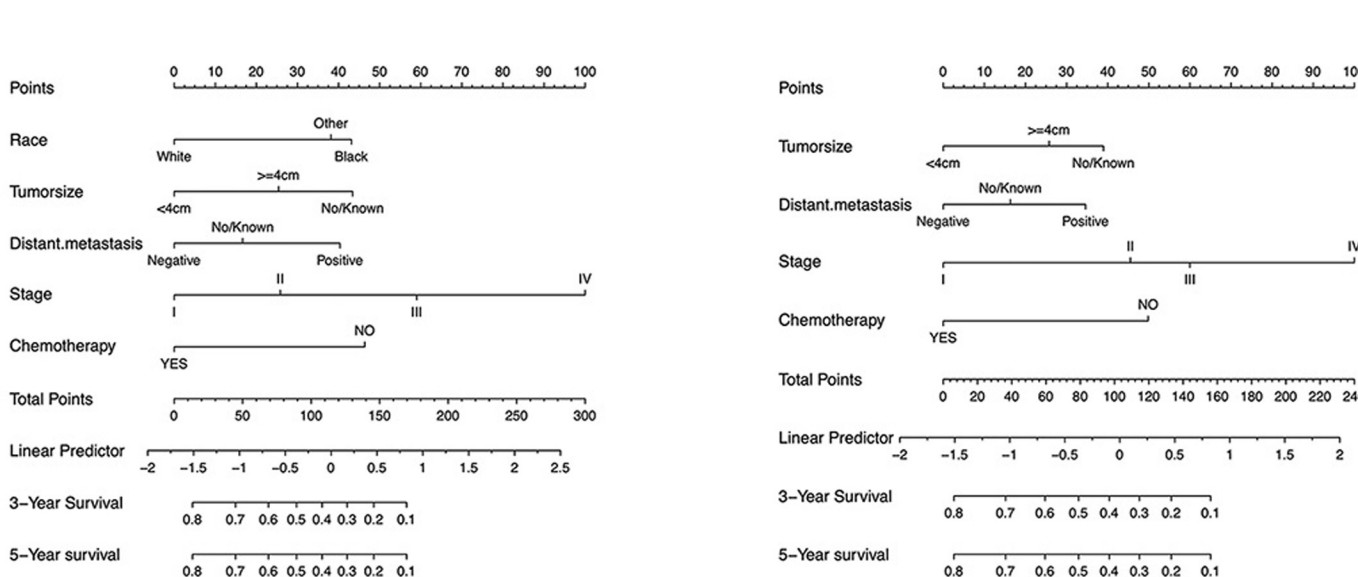

**Fig 2. Nomograms predicting the 3-year and 5-year overall survival of NECC patients.** Adding the scores of each independent prognostic factor, the overall survival was estimated by the total number of points for each factor on the bottom scale. A, five factors model; B, four factors model.

### Risk assessment

Based on the nomogram, we successfully divided the patients from different cohorts into high- and low-risk groups. By plotting Kaplan-Meier survival curves, we found the division for the training cohort (Fig 5A), internal validation (Fig 5B), and external validation cohort (Fig 5C) all showed statistically significant in differences OS for patients (p-value<0.001), further demonstrating the accuracy of the prognostic model.

## Discussion

In the current study, we successfully established and validated a prognostic model based on the SEER database and a single-center retrospective study for patients with NECC, a rare but lethal subtype of carcinoma of the uterine cervix. In our novel nomogram, the combination of race, tumor size, distant metastasis, stage and chemotherapy or tumor size, distant metastasis, stage, and chemotherapy precisely predict overall survival, especially 3-year OS of NECC patients in training, internal and external cohort. In the external cohort at YCH, we also demonstrated patients who received ≥4 cycles of chemotherapy may improve the OS of patients with NECC as well.

Based on the SEER database, our study revealed that the survival rate of NECC is significantly lower than that of squamous carcinoma and adenocarcinoma of the cervix in both early and advanced-stage patients. In the SEER database, the 3-year and 5-year OS rates were 38.9% and 35.2%, respectively, while those rates at YCH were 66.4% and 35.7%, respectively. A large retrospective study based on National Cancer Database (NCDB) recruited 127,332 patients, including 1,896 (1.5%) with NECC from 1998–2006, and fund 5-year survival rates were 55.4% for patients with stage IB tumors, 24.4% for those with stage IIIB tumors and 4.1% for those with stage IVB NECC [22]. Due to its remarkably poor prognosis, effective risk stratification tools are in urgent need to better distinguish patients for high-risk or recurrence and death,

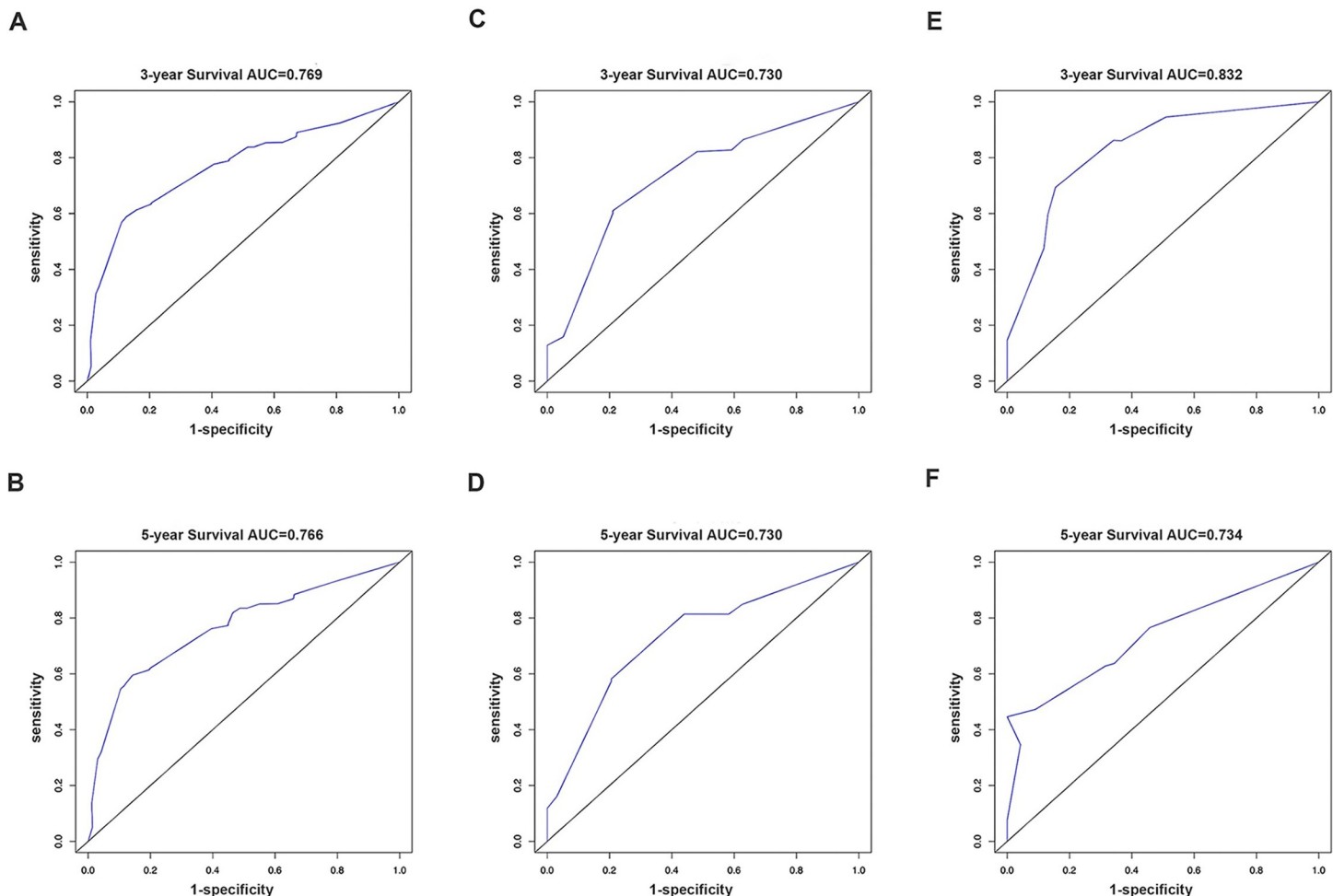

**Fig 3. 3-year overall survival AUC curves and 5-year overall survival AUC curves.** A and B, training cohort; C and D, internal validation cohort; E and F, external validation cohort.

then provide them with boost therapies. This study showed that race, tumor size, distant metastasis, stage, and chemotherapy posed thorough effects on overall survival based on the SEER database, consistent with previous studies extracted also from the SEER database [23–25]. Beyond that, we further constructed a visualized and practical model based on nomogram to better predict NECC patients' overall survival which yielded 0.742 (95% CI: 0.726–0.758) for C-index in the training cohort, 0.733 (95% CI: 0.709–0.757) in internal validation cohort and 0.792 (95% CI: 0.765–0.819) in external validation cohort (four factors model here, to extend the model for widely application of other cohorts with NECC), respectively. Besides, AUCs of the nomogram for predicting the 3-year and 5-year OS in the training cohort were 0.769 and 0.766, respectively, were 0.73 and 0.73 in the internal validation cohort, as well as 0.832 and 0.734 in the external validation cohort (four factors model). Both the C-index and AUCs confirmed the accuracy and effectiveness of the model we presented. Finally, we succeeded in dividing patients with NECC into high- and low-risk groups in the training group, internal validation, and external validation groups concerning OS using the four factors nomogram we developed.

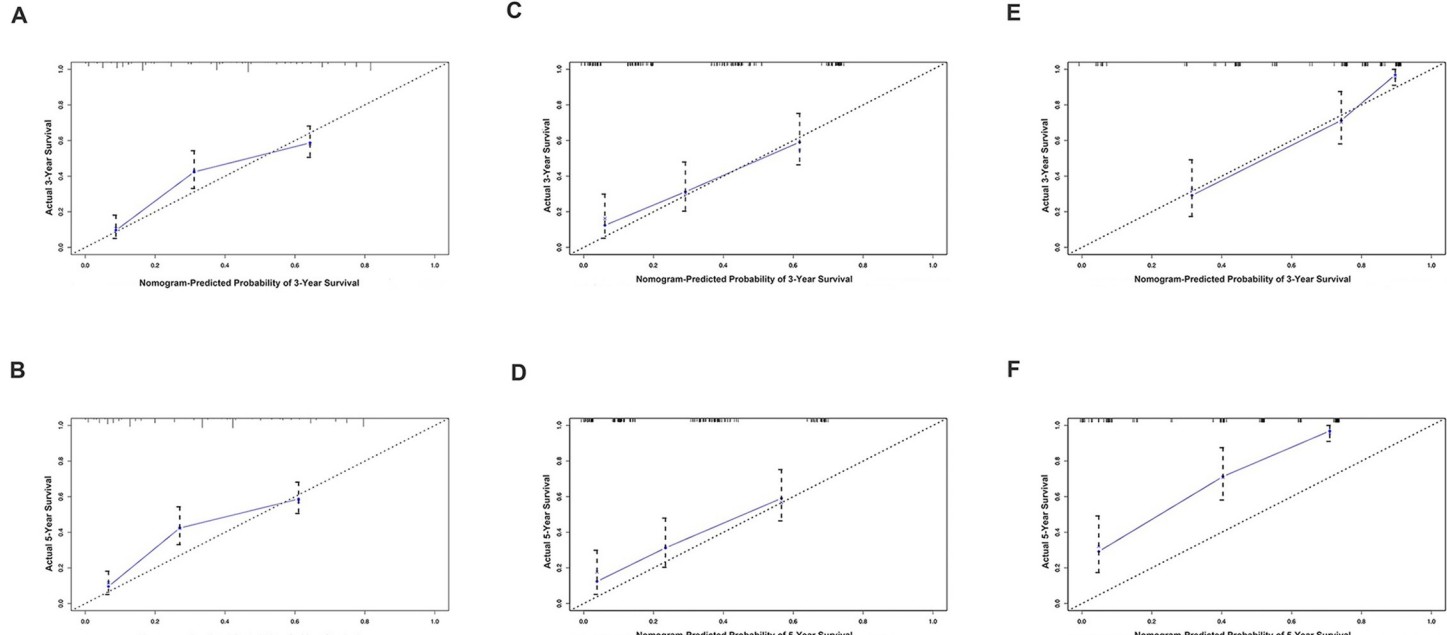

**Fig 4. 3-year overall survival calibration curves and 5-year overall survival calibration curves.** A and B, training cohort; C and D, internal validation cohort; E and F, external validation cohort.

In the external validation cohort of 122 patients at YCH, we also found tumor size, FIGO stage, depth of tumor stromal invasion, and distant metastasis were independent prognostic factors affecting patients' PFS, while tumor size, FIGO stage, depth of tumor stromal invasion and chemotherapy cycles were independent prognostic factors affecting patients' OS. Seino M et al found that for small cell neuroendocrine carcinoma of the cervix (SCNECC), tumors >4 cm in size had greater rates of lymph node and distant metastasis when compared with tumors </ = 4 cm [26]. Zhang X et al also confirmed advanced FIGO stage, tumor size > 4 cm, LNM, and LVSI were associated with poor survival for stage II SCNECC patients [27], while Gordhandas S et al revealed FIGO staging, rather than AJCC stage should be used to classify SCNECC [28]. Meta-analysis enrolled 20 studies funding FIGO staging, tumor size, parametrial involvement, resection margin, depth of stromal invasion, and LNM can be used as clinicopathological characteristics for the prediction of SCNECC prognosis [29]. A Chinese multicenter retrospective study suggested tumor diameter of >4 cm, LNM, DSI, and LVSI were confirmed as high-risk factors for worse DFS and OS in NECC patients. Patients with different numbers of risk factors had significantly different DFS and OS values [30]. In the current study, our prognostic model provided clinicians with an easy-to-use and quantitative tool to better scale the prognostic effect of those clinicopathologic factors on patients' survival. Our data from YCH also suggested more than 4 cycles of chemotherapy exerted a protective effect on patient's overall survival, while Wang R et al reported that post-operative chemotherapy alone showed no inferiority when compared with chemoradiotherapy for high-grade NECC and 4+ cycles of chemotherapy tended to produce a better prognosis than 4- ones [13]. In the cohort at YCH, the majority received surgery (102/122, 83.6%), mostly primary radical hysterectomy (79/102, 77.5%), while 22.5% (23/102) received neoadjuvant chemotherapy (NACT). A previous study showed surgery after NACT for locally advanced NECC may yield similar outcomes compared to concurrent chemoradiation (CRT) [31], which suggested radical surgery followed by chemotherapy may be a favorable alternative intervention for selected

**Table 3. Univariate and multivariate analysis of 3-year progression-free survival of NECC patients in the external validation cohort (N = 122).**

| Variables | | Univariate analysis | | | Multivariate analysis | | |
|---|---|---|---|---|---|---|---|
| | | 3-year PFS % | $\chi^2$ | P-value | HR | 95% CI | P-value |
| Age | <44 | 53.4 | 0.670 | 0.413 | | | |
| | ≥44 | 40.1 | | | | | |
| Menopausal status | No | 46.9 | 0.985 | 0.321 | | | |
| | Yes | 51.3 | | | | | |
| Serum NSE level | <16.3 | 53.3 | 11.465 | **0.001***  | reference | | 0.423 |
| | ≥16.3 | 31.4 | | | 1.298 | 0.686–2.459 | |
| Histology | Pure | 46.2 | 0.185 | 0.667 | | | |
| | Mixed | 53.8 | | | | | |
| Tumor size (cm) | <4 | 57.2 | 14.709 | **<0.001***  | reference | | 0.022*  |
| | ≥4 | 30.3 | | | 2.016 | 1.108–3.671 | |
| Stromal Invasion | Superficial 1/3 | 77.7 | 16.503 | **0.001***  | reference | | 0.008*  |
| | Middle 1/3 | 53.5 | | | 2.833 | 0.816–9.835 | 0.092 |
| | Deep 1/3 | 39.6 | | | 6.108 | 1.775–21.012 | 0.006 |
| | Unknown | 25.4 | | | 4.804 | 1.226–18.818 | 0.058 |
| LVSI | No | 57.7 | 7.932 | **0.019***  | reference | | 0.456 |
| | Yes | 21.5 | | | 0.901 | 0.420–1.930 | 0.668 |
| | Unknown | 25.4 | | | 1.630 | 0.732–3.629 | 0.053 |
| Lymph node metastasis | No | 60.4 | 10.781 | 0.005*  | | | |
| | Yes | 36.1 | | | | | |
| | Unknown | 25.4 | | | | | |
| Distant metastasis | No | 49.2 | 8.233 | **0.004***  | reference | | 0.253 |
| | Yes | 14.3 | | | 1.471 | 0.844–4.997 | |
| Stage | I | 76.3 | 32.060 | **<0.001***  | reference | | <0.001*  |
| | II | 34.8 | | | 2.501 | 1.127–5.549 | 0.024 |
| | III | 34.2 | | | 1.462 | 0.710–3.012 | 0.302 |
| | IV | 8.3 | | | 14.271 | 4.749–42.888 | <0.001 |
| Neoadjuvant chemotherapy | No | 47.5 | 1.385 | 0.239 | | | |
| | Yes | 60.9 | | | | | |
| Chemotherapy (Primary/Adjuvant only) | No | 38.8 | 1.489 | 0.222 | | | |
| | Yes | 48.2 | | | | | |
| Chemotherapy cycles | <4 | 38.1 | 2.431 | 0.119 | | | |
| | ≥4 | 54.6 | | | | | |
| Radiation | No | 53.5 | 0.403 | 0.526 | | | |
| | Yes | 42.6 | | | | | |

Abbreviations: PFS, Progression-Free Survival; HR, Hazard Ratio; CI, Confidence Interval; LVSI, lymph vascular space invasion.

* P-value <0.05. Border character indicating parameters included in multivariate analysis.

patients with advanced-stage cancer [14]. It is also worth noting that in the current study, in both the SEER cohort and YCH cohort, radiation failed to demonstrate benefit concerning OS for patients with NECC, and our recent multicenter, retrospective cohort study enrolled more than 1,000 patients suggesting surgery may be superior to radiation for SCNECC patients, especially for those at locally advanced stages [32]. Thus, the primary treatment algorithm for patients with NECC may need further optimization based on gradually emerging new evidence.

**Table 4. Univariate and multivariate analysis of 3-year overall survival of NECC patients in the external validation cohort (N = 122).**

| Variables | | Univariate analysis | | | Multivariate analysis | | |
|---|---|---|---|---|---|---|---|
| | | 3-year OS % | $\chi^2$ | P-value | HR | 95% CI | P-value |
| Age | <44 | 68.7 | 0.736 | 0.391 | | | |
| | ≥44 | 64.9 | | | | | |
| Menopausal status | No | 68.8 | 0.166 | 0.684 | | | |
| | Yes | 65.9 | | | | | |
| Serum NSE level | <16.3 | 73.2 | 9.245 | **0.002**[*] | reference | | 0.217 |
| | ≥16.3 | 49.1 | | | 0.637 | 0.312–1.303 | |
| Histology | Pure | 66.4 | 0.167 | 0.683 | | | |
| | Mixed | 67.3 | | | | | |
| Tumor size (cm) | <4 | 80.8 | 28.493 | **<0.001**[*] | reference | | 0.005[*] |
| | ≥4 | 36.7 | | | 2.682 | 1.348–5.336 | |
| Stromal Invasion | Superficial 1/3 | 80.9 | 14.955 | **0.001**[*] | reference | | 0.005[*] |
| | Middle 1/3 | 78.7 | | | 2.493 | 0.547–11.364 | 0.068 |
| | Deep 1/3 | 56.3 | | | 5.994 | 1.360–26.421 | 0.001 |
| | Unknown | 44.4 | | | 2.269 | 0.477–10.799 | 0.019 |
| LVSI | No | 75.6 | 10.908 | **0.004**[*] | reference | | 0.233 |
| | Yes | 47.4 | | | 1.550 | 0.663–3.630 | 0.261 |
| | Unknown | 44.4 | | | 3.315 | 0.659–16.673 | 0.099 |
| Lymph node metastasis | No | 77.0 | 17.489 | **<0.001**[*] | | | |
| | Yes | 52.1 | | | | | |
| | Unknown | 44.4 | | | | | |
| Distant metastasis | No | 69.9 | 16.877 | **<0.001**[*] | reference | | 0.025[*] |
| | Yes | 0.0 | | | 1.806 | 1.119–2.876 | |
| Stage | I | 79.4 | 54.796 | **<0.001**[*] | reference | | 0.005[*] |
| | II | 63.4 | | | 2.606 | 0.970–7.007 | 0.058 |
| | III | 55.3 | | | 2.426 | 1.015–5.800 | 0.046 |
| | IV | 8.3 | | | 12.921 | 4.734–35.266 | <0.001 |
| Neoadjuvant chemotherapy | No | 71.5 | 0.019 | 0.889 | | | |
| | Yes | 64.3 | | | | | |
| Chemotherapy (Primary/Adjuvant only) | No | 66.3 | 0.935 | 0.334 | | | |
| | Yes | 67.4 | | | | | |
| Chemotherapy cycles | <4 | 56.1 | 11.808 | **0.001**[*] | reference | | 0.035[*] |
| | ≥4 | 75.6 | | | 0.504 | 0.267–0.953 | |
| Radiation | No | 67.4 | 0.001 | 0.972 | | | |
| | Yes | 65.5 | | | | | |

Abbreviations: PFS, Progression-Free Survival; HR, Hazard Ratio; CI, Confidence Interval; LVSI, lymph vascular space invasion.

[*] P-value <0.05. Border character indicating parameters included in multivariate analysis.

Previous studies revealed the positive rates of neuroendocrine immunohistochemistry markers in NECC and found the positive rates of CGA, SYN, and CD56 were high, and NSE was a moderately sensitive index. P16 and Ki67 were the most sensitive [33], while Syn and CD56 are reliable indicators for diagnosing SCNECC [34]. In this study, we found Syn negativity could lead to a worse prognosis, but we did not confirm it in the multivariate analysis due to the large difference in the number of negative and positive results in the postoperative pathology, which tends to bias the results. No other immunological indicators were found to

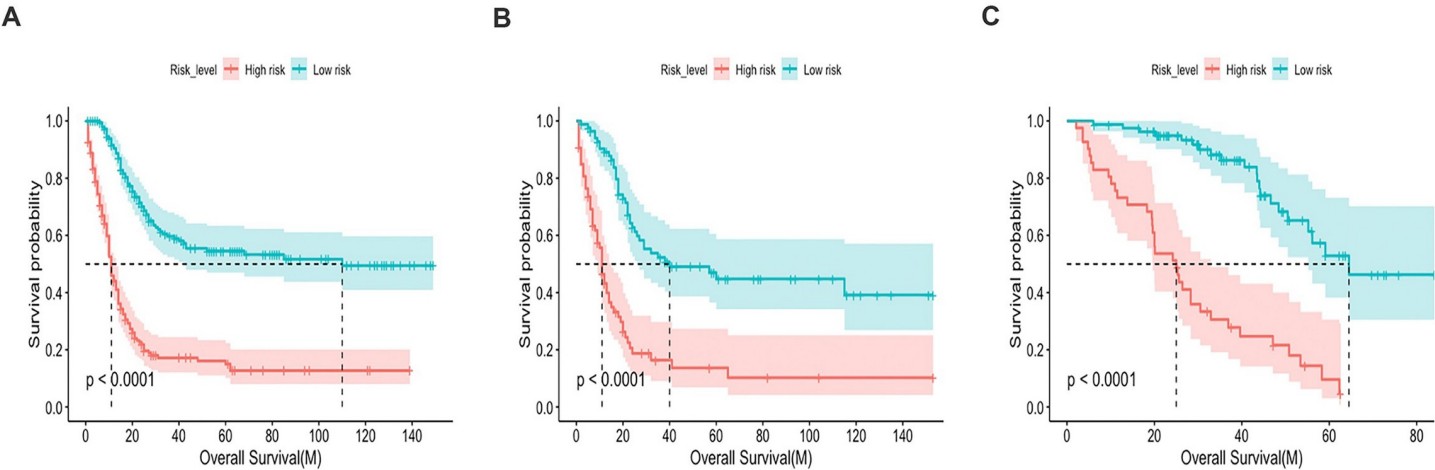

**Fig 5. Comparison of overall survival between high- and low-risk groups.** A, training cohort; B, internal validation cohort; C, external validation cohort.

have an impact on prognosis, which suggested the prognostic value of neuroendocrine immunohistochemistry markers in NECC demands further investigation.

The current study was limited by its retrospective nature, and our model in the training cohort showed that the predicted 5-year rates were far beyond the actual ones. Moreover, the 5-year OS of the YCH cohort could not be validated because there was a limited number of patients with a follow-up time of at least 5 years. In addition, this rare disease requires a lengthy period for cohort recruiting and often there is incomplete information from early subjects, leading to some unsatisfactory results.

Since several pre-clinical studies suggested that patients with NECC may benefit from immune checkpoint inhibitors (ICIs) [35–37], while other case reports showed promising results from ICIs therapy [38, 39], we are planning a single-arm phase two study to investigate the safety as well as efficacy of ICIs for patients with recurrent NECC, and clinical trials with ICIs designed for patients with high-risk of recurrence and death distinguished by our model presented in the current study are under investigation.

In conclusion, this study successfully established and validated a simple yet redactable prognostic model that might assist clinical judgment and treatment of patients with NECC for the first time and laid the foundation for subsequent studies and even prospective clinical trials.

## Supporting information

**S1 Fig. Comparison of survival analysis of different clinicopathological factors affecting OS in 376 patients with NECC in the training cohort.** A, Race; B, Tumor size; C, Lymph node metastasis; D, Distant metastasis; E, Stage; F, Chemotherapy.
(TIF)

**S2 Fig. Comparison of survival analysis of different clinicopathological factors affecting PFS in 122 patients with NECC in the external validation cohort.** A, Serum NSE; B, Tumor size; C, Stage; D, Stromal Invasion; E, Lymph node metastasis; F, Distant metastasis.
(TIF)

**S3 Fig. Comparison of survival analysis of different clinicopathological factors affecting OS in 122 patients with NECC in the external validation cohort.** A, Serum NSE; B. Tumor

size; C, Stage; D, Stromal Invasion; E, Lymph node metastasis; F, Cycles of chemotherapy.
(TIF)

**S1 Table. The clinicopathologic characteristics of NECC patients in the external validation cohort.**
(DOCX)

**S2 Table. Univariate analysis of immunohistochemical markers of 3-year progression-free survival and 3-year overall survival of NECC patients in the external validation cohort (N = 122).** * P-value <0.05.
(DOCX)

## Acknowledgments

The authors express gratitude to all study participants and other research staff (Innovation Experiment for College Students at Kunming Medical University in 2021, 2021JXD083, Yanchen Liu, Yajiao Duan, Yisi Huang, Jingxia Yan, Shuangli Yu) who participated in the work.

## Author Contributions

**Conceptualization:** Hongying Yang.

**Data curation:** Caixian Yu, Shao Zhang, Hongying Yang.

**Formal analysis:** Hongying Yang.

**Investigation:** Xiaoliu Wu, Shao Zhang, Min Zhao.

**Methodology:** Caixian Yu, Shao Zhang, Lan Zhang, Hongping Zhang, Min Zhao.

**Project administration:** Xiaoliu Wu, Shao Zhang, Lan Zhang, Hongping Zhang, Min Zhao, Zheng Li.

**Resources:** Hongping Zhang, Zheng Li.

**Software:** Caixian Yu, Hongping Zhang.

**Supervision:** Lan Zhang, Hongying Yang, Zheng Li.

**Validation:** Zheng Li.

**Visualization:** Zheng Li.

**Writing – original draft:** Caixian Yu.

**Writing – review & editing:** Zheng Li.

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
