## [Decision Letter · Decision Letter 0]

17 Oct 2023

PONE-D-23-17611Prognostic model for survival in patients with neuroendocrine carcinomas of the cervix: SEER database analysis and a single-center retrospective studyPLOS ONE

Dear Dr. Li,

Thank you for submitting your manuscript to PLOS ONE. After careful consideration, we feel that it has merit but does not fully meet PLOS ONE’s publication criteria as it currently stands. Therefore, we invite you to submit a revised version of the manuscript that addresses the points raised during the review process.

We look forward to receiving your revised manuscript.

Kind regards,

Andrea Giannini

Academic Editor

PLOS ONE

2. In your cover letter, please indicate whether you received any third party support in conducting this research, analyzing the data, or preparing the manuscript for submission. If yes, provide details as to the organization(s) involved and their specific contributions.

Additional Editor Comments:

Dear authors,

the topic of the present article titled “Prognostic model for survival in patients with neuroendocrine carcinomas of the cervix: SEER database analysis and a single-center retrospective study” is very interesting, the paper and the aim falls within the scope of the journal but the article needs major improvements.

The introduction, material and method section and tables should be modified and improved.

The manuscript should be organized better and English should be improved.

I suggest improving the manuscript with the reviewers' comments.

Reviewers' comments:

Reviewer's Responses to Questions

**Comments to the Author**

1. Is the manuscript technically sound, and do the data support the conclusions?

Reviewer #1: Yes

Reviewer #2: Yes

Reviewer #3: Yes

Reviewer #4: Partly

2. Has the statistical analysis been performed appropriately and rigorously? 

Reviewer #1: Yes

Reviewer #2: Yes

Reviewer #3: Yes

Reviewer #4: Yes

3. Have the authors made all data underlying the findings in their manuscript fully available?

Reviewer #1: Yes

Reviewer #2: Yes

Reviewer #3: Yes

Reviewer #4: Yes

4. Is the manuscript presented in an intelligible fashion and written in standard English?

Reviewer #1: Yes

Reviewer #2: Yes

Reviewer #3: Yes

Reviewer #4: Yes

5. Review Comments to the Author

Reviewer #1: In this manuscript, the authors successfully established and validated a prognostic model based on SEER data base

for patients with NECC. The retrospective study for patients with NECC also be processed. Retrospective studies have also well validated this prognostic model.

Reviewer #2: The present article seems precise and reliable using SEER data and medical information from the cases at YCH. As you mentioned in this manuscript, NECC is extremely rare with poor prognosis. In such types of diseases only restricted information is available even in the multicenter studies. We may get the great insight to the rare types of uterine cervical cancer through this article.

In this manuscript some points which should be clarify were detected.

1) In line 191 to 192, you mentioned the cumulative 3-year progression-free rate was ‘3.2 %’, which was fewer than the 5-year rate, 35.6 %. Please check the number.

2) In line 275, you should correct the word ‘fund’ to ‘found’.

3) In line 254 and 256, SNCEC should be correct to SCNEC.

4) In Table 3, 4, there are ‘chemotherapy’ and ‘Chemotherapy’ in the Variables, which make the readers confused. If possible, it may be better to add some annotations.

5) Not only internal validation cohort using SEER data, but also external validation cohort using YCH data, there are no merits to treat using radiotherapy for the patients suffering from NECC cervical canecr. In general radiotherapy is useful and helps to improve the prognosis for the patients suffering from cervical cancer, however, it is different for the NECC cervical cancer. Although you mentioned these comments in discussion, you should announce the fact using strong words.

Reviewer #3: I read with great interest the manuscript, which falls within the aim of this Journal and offers a high-quality overview of the topic.

Although the manuscript can be considered already of high quality, I would suggest to take into account the following minor recommendations:

- I suggest another round of language revision, in order to correct few typos and improve readability.

- I find it interesting to include a reference to the screening programs for early cervical cancer diagnosis (Golia D'Augè T, Giannini A, Bogani G, Di Dio C, Laganà AS, Di Donato V, Salerno MG, Caserta D, Chiantera V, Vizza E, Muzii L, D’Oria O. Prevention, Screening, Treatment and Follow-Up of Gynecological Cancers: State of Art and Future Perspectives. Clin. Exp. Obstet. Gynecol. 2023, 50(8), 160. https://doi.org/10.31083/j.ceog5008160).

- The authors have not adequately highlighted the strengths of their study. I suggest better specifying these points.

Reviewer #4: This is a relatively straightforward "data-analysis" type of manuscript, where the goal is to consider data from NECC patients available in SEER, and construct nomograms for efficient survival prediction. On the overall, I enjoyed the analysis steps. I do have some questions:

1. ON what basis was a 7:3 (training: validation) selection made?

2. Cox proportional hazards model (and coxph() function) was used for nomogram constructions; how was the proportionality assumption assessed?

3. The variable selection carried out in Results (Subsection 2) appeared somewhat adhoc; any reason, why any formal method (like Lasso, etc) were not used?

6. PLOS authors have the option to publish the peer review history of their article (what does this mean?). If published, this will include your full peer review and any attached files.

Reviewer #1: No

Reviewer #2: No

Reviewer #3: **Yes: **Ottavia D'Oria

Reviewer #4: No

---

## [Author Response · Author response to Decision Letter 0]

2 Nov 2023

Reviewer: 1

In this manuscript, the authors successfully established and validated a prognostic model based on SEER data base for patients with NECC. The retrospective study for patients with NECC also be processed. Retrospective studies have also well validated this prognostic model.

Response：We appreciate the reviewer’s positive comments on our manuscript.

Reviewer: 2

1) In line 191 to 192, you mentioned the cumulative 3-year progression-free rate was ‘3.2 %’, which was fewer than the 5-year rate, 35.6 %. Please check the number.

Response: We appreciate the reviewer’s carefully checking. We checked the raw data and found out the cumulative progression-free survival rate of 3 years at YCH should be 53.2%. We believe it was a typo and have corrected it in the revised manuscript.

2) In line 275, you should correct the word ‘fund’ to ‘found’.

Response: We have made modifications in the corresponding positions in the revised manuscript as the reviewer suggested. Thanks for pointing out another typo.

3) In line 254 and 256, SNCEC should be correct to SCNEC.

Response: Yes, it should be “SCNECC” instead of “SNCEC,” and we have made the correction accordingly for all these acronyms in the revised manuscript.

4) In Table 3, 4, there are ‘chemotherapy’ and ‘Chemotherapy’ in the Variables, which make the readers confused. If possible, it may be better to add some annotations.

Response: We use “Chemotherapy” in Table 3 and 4 in the Variables lines to make it more clearly in the revised manuscript.

5) Not only internal validation cohort using SEER data, but also external validation cohort using YCH data, there are no merits to treat using radiotherapy for the patients suffering from NECC cervical canecr. In general radiotherapy is useful and helps to improve the prognosis for the patients suffering from cervical cancer, however, it is different for the NECC cervical cancer. Although you mentioned these comments in discussion, you should announce the fact using strong words.

Response: Yes, and our latest multi-centers retrospective study published on Lancet Oncology confirmed that even for patients with local advanced stage of NECC, they are more likely benefit from surgery rather than radiation, and we added a short graph in discussion part to emphasize that idea with a new reference [30]. (line in revised manuscript).

Reviewer: 3

I read with great interest the manuscript, which falls within the aim of this Journal and offers a high-quality overview of the topic.

Although the manuscript can be considered already of high quality, I would suggest to take into account the following minor recommendations:

- I suggest another round of language revision, in order to correct few typos and improve readability.

Response: We believe we have corrected all the typos and gramma errors, and we rewrote the introduction, as well as material and method section in the revised manuscript to improve its readability.

- I find it interesting to include a reference to the screening programs for early cervical cancer diagnosis (Golia D'Augè T, Giannini A, Bogani G, Di Dio C, Laganà AS, Di Donato V, Salerno MG, Caserta D, Chiantera V, Vizza E, Muzii L, D’Oria O. Prevention, Screening, Treatment and Follow-Up of Gynecological Cancers: State of Art and Future Perspectives. Clin. Exp. Obstet. Gynecol. 2023, 50(8), 160. https://doi.org/10.31083/j.ceog5008160).

Response: We really appreciate that idea and believe that is a decent paper. But after careful consideration, we finally decided not to add this review as a new reference since this manuscript focused only on the establishment and confirmation of prognostic model for survival in patients with NECC, and we will cite this review on our other papers concerning the screening programs of cervical cancer. 

- The authors have not adequately highlighted the strengths of their study. I suggest better specifying these points.

Response: We highlighted the strengths of our study in the revised introduction and discussion sections in the revised manuscript, thanks for the remind. 

Reviewer: 4

1） ON what basis was a 7:3 (training: validation) selection made?

Response: As a matter of fact, there is no consensus on the optimal partition ratio of training set versus validation set in the establishment of prediction model till now. In previous studies, the partition ratio of the training set and validation set ranges from 5:5 to 7:3. However, the more accurate the model is, the larger sample size it requires. Taken that the accuracy of the model we established is about 98%, a ratio of 5:5 will require at least 3,000 patients in each set, and that will not be practical in the current study. That is why we finally chose a partition ratio of 7:3 to assign more participants in the training set to improve the model’s accuracy in a relatively small sample size, and we added two more extra reference to demonstrate how that choice are made (reference 19, 20).

2） Cox proportional hazards model (and coxph() function) was used for nomogram constructions; how was the proportionality assumption assessed?

Response: In the current study, five clinicopathologic factors (race, tumor size, distant metastasis, stage, and chemotherapy) were selected by multivariate analyses based on Cox proportional hazards model and then were enrolled in nomogram construction. As far as we know, all the clinicopathologic factors mentioned above (except race）are graded variables and Cox proportional hazards model was the most common and reasonable tool to analyze the impact those graded variables on survival by a dose-dependent manner (for example, tumor size, HR=1.445; distant metastasis, HR=1.804; chemotherapy, HR=0.513), as most previous studies have been done (reference 5-9). Then the nomogram (especially the four factors one excluding race) we constructed is also based on the proportionality assumption, because the variables that built the nomogram are all graded variables, so they exert their impact on survival on a dose-dependent manner as we mentioned before, and we believe that is why other previous studies made the same proportionality assumption when they built their nomograms in their works (reference 17, 18). 

3） The variable selection carried out in Results (Subsection 2) appeared somewhat adhoc; any reason, why any formal method (like Lasso, etc) were not used?

Response: Thanks for your kindly and professional advice concerning statistic processes. We believe that the selection of variable in medical studies like the current one, is based on both scientific reasons and statistical ones. In this particular case, as we mentioned in “Univariate and multivariate analysis in the training cohort” (Results, subsection 2), age, race, tumor size, distant metastasis, stage, LNM and chemotherapy were variables that were selected through univariate analysis, but we decided not to put LNM in the multivariate analysis because the variable “stage” (AJCC stage but not FIGO stage in the training set) here, is actually containing the information of LNM, and that explained why a strong collinear relationship between LNM and stage were observed. Furthermore, there were 62.5% of patients in training cohort (235/376) whose LNM statuses were marked as “unknown”, and any statistical process based on such a high deletion rate is risky. So, we decided not to enroll “LNM” in the multivariate analysis but keep “stage” in since the latter contains more information rather than the former. Furthermore, we usually employed Lasso in multivariate analysis when there were too many variables (usually 10 or more) stood from univariate analysis and we kept deleting variables for scientific reasons. We also rewrote Results, subsection 2 in the revised manuscript to explain our selection of variables more clearly, hopefully that will not cause any ad hoc feelings for our reviewers and readers anymore.

---

## [Decision Letter · Decision Letter 1]

21 Nov 2023

PONE-D-23-17611R1Prognostic model for survival in patients with neuroendocrine carcinomas of the cervix: SEER database analysis and a single-center retrospective studyPLOS ONE

Dear Dr. Li,

Thank you for submitting your manuscript to PLOS ONE. After careful consideration, we feel that it has merit but does not fully meet PLOS ONE’s publication criteria as it currently stands. Therefore, we invite you to submit a revised version of the manuscript that addresses the points raised during the review process.

We look forward to receiving your revised manuscript.

Kind regards,

Andrea Giannini

Academic Editor

PLOS ONE

Journal Requirements:

Additional Editor Comments:

Dear authors,

the manuscript it has now been evaluated by our experts and they have recommended that minor changes be made to the submission.

Reviewers' comments:

Reviewer's Responses to Questions

**Comments to the Author**

1. If the authors have adequately addressed your comments raised in a previous round of review and you feel that this manuscript is now acceptable for publication, you may indicate that here to bypass the “Comments to the Author” section, enter your conflict of interest statement in the “Confidential to Editor” section, and submit your "Accept" recommendation.

Reviewer #2: All comments have been addressed

Reviewer #4: (No Response)

2. Is the manuscript technically sound, and do the data support the conclusions?

Reviewer #2: Yes

Reviewer #4: Yes

3. Has the statistical analysis been performed appropriately and rigorously? 

Reviewer #2: Yes

Reviewer #4: No

4. Have the authors made all data underlying the findings in their manuscript fully available?

Reviewer #2: Yes

Reviewer #4: Yes

5. Is the manuscript presented in an intelligible fashion and written in standard English?

Reviewer #2: Yes

Reviewer #4: Yes

6. Review Comments to the Author

Reviewer #2: This revised manuscript seems well-corrected enough to be published.

I am so satisfied with your great job.

Reviewer #4: The authors addressed my previous comments, with a great degree of satisfaction. However, the reply on "checking proportional hazards assumptions" was not adequate. Notethat, there are formal tests to assess the assumption; see here: https://stat.ethz.ch/R-manual/R-devel/library/survival/html/cox.zph.html

I understand that failure of those assumptions may lead to exploring a different model, suich as the accelerated failure time, etc, which may not have the elegant PH interpretations (in terms of hazard ratios). To provide some statistical rigor to this manuscript, the authors are advised either to formally check the assumptions. Even if the assumptions fail, they may comment that for the sake of interpretability, they are sticking to the Cox PH model, but future research may pertain to exploring alternative modeling.

7. PLOS authors have the option to publish the peer review history of their article (what does this mean?). If published, this will include your full peer review and any attached files.

Reviewer #2: No

Reviewer #4: No

---

## [Author Response · Author response to Decision Letter 1]

28 Nov 2023

Reviewer #4: The authors addressed my previous comments, with a great degree of satisfaction. However, the reply on "checking proportional hazards assumptions" was not adequate. Notethat, there are formal tests to assess the assumption; see here: https://stat.ethz.ch/R-manual/R-devel/library/survival/html/cox.zph.html

I understand that failure of those assumptions may lead to exploring a different model, suich as the accelerated failure time, etc, which may not have the elegant PH interpretations (in terms of hazard ratios). To provide some statistical rigor to this manuscript, the authors are advised either to formally check the assumptions. Even if the assumptions fail, they may comment that for the sake of interpretability, they are sticking to the Cox PH model, but future research may pertain to exploring alternative modeling

Response: Thank you for your feedback and for providing the link to the formal tests for checking proportional hazards (PH) assumptions. We apologize for not adequately addressing this concern in our previous response, and we understand the importance of assessing the assumptions and the potential implications for model interpretation. Considering your suggestion, we consulted our expert on statistics (Pro. Min Zhao) and employed two methods (Global test and Schoenfeld residual graph) to check the assumptions. Firstly, we run the Cox.zph function in the R language survival package for Global test. If the P-values of each covariate are greater than 0.05, it indicates that each variable meets the PH test, and while the overall test P-value of the model is also greater than 0.05, it suggests the model meets the PH test.

Here are our output results:

 chisq df P-value

Race 0.0829 1 0.773

Stage 3.1172 1 0.077

Tumorsize 2.8043 1 0.094

Distant.metastasis 0.5762 1 0.448

Chemotherapy 0.1177 1 0.732

GLOBAL 6.2295 5 0.285

Now we can find out that each covariate, as well as the global test, is of P-value greater than 0.05, which means that the Cox model conforms to PH assumption.

Then we used the ggcoxzph() function to generate a normalized Schoenfeld residual correlation image concerning time for each covariate, and the output image is printed in the file "Respond to Reviewers". 

In the above figures, solid lines represent the smooth spline fitting of the curve, while the dashed line represents the+/-2 times standard error range around the fitting. Because we assume that the coefficient β 1, β 2, and β 3 does not change over time, from the graph inspection, the covariates we included vary over time, which means all covariates satisfy the assumption of PH.

To sum up, we employed two formal tests as the reviewer indicated to check whether our model met the proportional hazards (PH) assumptions and it turns out the model we built did meet the assumption. So, we added a short sentence in the “Statistical methods” part in the current revised manuscript (lines 132 to 134) to indicate that idea. 

Thank you again for bringing this to our attention, and we will ensure that the manuscript includes a more thorough discussion on checking proportional hazards assumptions and the potential implications for future research.

---

## [Decision Letter · Decision Letter 2]

13 Dec 2023

Prognostic model for survival in patients with neuroendocrine carcinomas of the cervix: SEER database analysis and a single-center retrospective study

PONE-D-23-17611R2

Dear Dr. Li,

We’re pleased to inform you that your manuscript has been judged scientifically suitable for publication and will be formally accepted for publication once it meets all outstanding technical requirements.

Kind regards,

Andrea Giannini

Academic Editor

PLOS ONE

Additional Editor Comments (optional):

The manuscript has been modified with the comments of the reviewers. It is now ready to be published.

Reviewers' comments:

Reviewer's Responses to Questions

**Comments to the Author**

1. If the authors have adequately addressed your comments raised in a previous round of review and you feel that this manuscript is now acceptable for publication, you may indicate that here to bypass the “Comments to the Author” section, enter your conflict of interest statement in the “Confidential to Editor” section, and submit your "Accept" recommendation.

Reviewer #2: All comments have been addressed

Reviewer #4: All comments have been addressed

2. Is the manuscript technically sound, and do the data support the conclusions?

Reviewer #2: Yes

Reviewer #4: (No Response)

3. Has the statistical analysis been performed appropriately and rigorously? 

Reviewer #2: Yes

Reviewer #4: (No Response)

4. Have the authors made all data underlying the findings in their manuscript fully available?

Reviewer #2: Yes

Reviewer #4: (No Response)

5. Is the manuscript presented in an intelligible fashion and written in standard English?

Reviewer #2: Yes

Reviewer #4: (No Response)

6. Review Comments to the Author

Reviewer #2: I am very pleased with the second-revised manuscript. Over and over the authors revise the manuscript, it becomes more and more sophisticated. It seems that this manuscript has reached to be accepted.

Reviewer #4: (No Response)

7. PLOS authors have the option to publish the peer review history of their article (what does this mean?). If published, this will include your full peer review and any attached files.

Reviewer #2: No

Reviewer #4: No

---

## [Editor Report · Acceptance letter]

26 Dec 2023

PONE-D-23-17611R2 

PLOS ONE

Dear Dr. Li, 

I'm pleased to inform you that your manuscript has been deemed suitable for publication in PLOS ONE. Congratulations! Your manuscript is now being handed over to our production team.

Kind regards, 

on behalf of

Dr. Andrea Giannini 

Academic Editor

PLOS ONE